# Comparative effectiveness of non-pharmacological therapies for postoperative cognitive dysfunction: Protocol for a systematic review and network meta-analysis

Kexin Wu[1,2☉], Qiongnan Bao[3☉], Jun Huang[1☉], Shanshan Sun[4], Yaqin Li[1], Xinyue Zhang[1,2], Manze Xia[1,2], Zhenghong Chen[1,2], Jin Yao[1,2], Wanqi Zhong[1,2], Zihan Yin[1,2]*, Fanrong Liang[1,2]*

**1** Acupuncture and Tuina School, Chengdu University of Traditional Chinese Medicine, Chengdu, China, **2** Sichuan Provincial Acupuncture Clinical Research Center, Chengdu, China, **3** Department of traditional Chinese medicine, The First People's Hospital of Yunnan Province, The Affiliated Hospital of Kunming University of Science and Technology, Kunming, China, **4** First Teaching Hospital of Tianjin University Chinese Medicine, Tianjin, China

☉ These authors contributed equally to this work.
* acuresearch@126.com (FL); yinzihan@stu.cdutcm.edu.cn (ZY)

**Data Availability Statement:** No datasets were generated or analysed during the current study. All

# Abstract

## Introduction

Postoperative cognitive dysfunction (POCD) is a common complication following surgery. Electroacupuncture (EA), manual acupuncture (MA), transcutaneous electrical acupoint stimulation (TEAS), and cognitive training (CT) can effectively maintain or improve the post-operative cognitive function of patients. However, it remains unclear which therapy is the most effective. Therefore, this network meta-analysis aims to compare and rank the efficacy of these non-pharmacological therapies for POCD to identify the optimal therapy.

## Methods and analysis

A systematic search will be conducted across seven databases (PubMed, Cochrane Library, EMBASE, Web of Science, CINAHL, AMED, and PsycINFO) for articles published between January 2000 and November 2023. Two reviewers will independently conduct study selection and data extraction. The primary outcome will be the changes in the overall cognitive function before and after the intervention. The secondary outcome will be the incidence of POCD. The risk of bias will be assessed using the revised Risk of Bias Assessment Tool. Pairwise and Bayesian network meta-analyses will be performed using RevMan, STATA, and Aggregate Data Drug Information System statistical software. Additionally, the quality of evidence will be assessed using the Grading of Recommendations Assessment, Development, and Evaluation guidelines.

**Ethics and dissemination:** The results will be disseminated to peer-reviewed journals or conferences.

relevant data from this study will be made available upon study completion.

**Funding:** This study received financial support from the State Administration of Traditional Chinese Medicine, under the National Key Research and Development Program of China (Grant No. 2019YFC1709700), the National Natural Science Foundation of China (Grants Nos. 81973961, 82004486), and the Project of the Science and Technology Department of Sichuan Province (Grant No. 2021YFS0087). Fanrong Liang acknowledges support from these organizations as a grantee. The funders had no role in study design, data collection and analysis, decision to publish, or preparation of the manuscript.

**Competing interests:** The authors have declared that no competing interests exist.

## Trial registration

PROSPERO registration number: CRD42023454028.

## Introduction

Postoperative cognitive dysfunction (POCD) is a common postoperative complication of the central nervous system, defined as cognitive decline occurring within one year after surgery [1, 2]. This decline is characterised by a decrease in memory, abstraction, and attention [3]. The incidence of POCD varies between 1.5% and 28% [4]. Moreover, with the growing global ageing population increases and advancements in medical technology, the number of patients undergoing surgery will continue to increase, along with the risk of developing POCD [5]. POCD can lead to consequences such as delayed postoperative recovery, low quality of life, early retirement, and increased mortality [3]. A recent study demonstrated that patients with postoperative neurocognitive disorders face an increased mortality of 10.2% and an increase of $17,275 in costs one year after hospital admission [3, 6]. This poses a huge burden on patients, families, and society. Therefore, POCD is an issue that needs urgent attention and solutions.

Currently, there are no clinical guidelines for POCD. Common clinical prevention and treatment methods include anaesthesia management [7], perioperative nursing [8], and intraoperative monitoring [9, 10]. Nevertheless, limited clinical improvements in postoperative cognitive ability have been identified to date. Various drugs, such as probiotic treatment [11], dexmedetomidine [12, 13], donepezil [14], and benzodiazepines [4], are used for cognitive impairment in the clinic. However, their effectiveness and potential adverse effects in POCD remain unclear owing to a lack of high-quality clinical trials. Therefore, it is necessary to explore more effective and safe methods to prevent and treat POCD.

Non-pharmacological therapies for POCD are gaining increasing attention, with four approaches being the most commonly studied. In recent years, many random controlled trials (RCTs) have demonstrated that electroacupuncture (EA) [15], manual acupuncture (MA) [16], transcutaneous electrical acupoint stimulation (TEAS) [17], and cognitive training (CT) [18] exhibit positive effects on the cognitive function of postoperative patients. MA is a traditional Chinese medicine technique that involves inserting thin needles into specific points on the body, known as acupuncture points or acupoints. EA is a modification of traditional MA where a small electrical current is passed between pairs of acupuncture needles. A systematic review (SR) published in 2023 [19] revealed that acupuncture-related techniques, including MA and EA, are associated with fewer postoperative cognitive complications. TEAS is a non-invasive method that uses electrodes to apply a small electrical current to the skin near acupuncture points [20]. An SR published in 2022 [21] indicated that TEAS can prevent early postoperative cognitive decline after general anaesthesia in older adults. Cognitive training involves the scientific assessment and systematic training of cognitive abilities. An SR and meta-analysis by Bowden et al. [22] indicated that cognitive intervention is effective in improving cognitive function after general surgery. Moreover, guidelines for other cognitive disorders [23] have recommended cognitive stimulation to improve cognitive function. Thus, non-pharmacological interventions are promising preventive and therapeutic strategies for POCD. Nevertheless, despite the effectiveness of these four methods, it remains unclear which method is the most effective.

Traditional meta-analysis can only directly compare the effects of two approaches, while network meta-analysis (NMA) allows for the comparison of multiple interventions by

combining direct and indirect evidence from different RCTs to identify the most effective intervention [24]. Thus, the present study will be the first to use NMA to compare the efficacy of four non-pharmacological interventions for POCD and highlight the most optimal intervention, providing a clinical reference for clinicians and patients.

## Methods and analysis

### Design and registration

The SR protocol was registered with PROSPERO (CRD42023454028). This review will be reported according to the Preferred Reporting Items for Systematic Review and Meta-Analysis NMA (PRISMA-NMA) [25] and its relevant extensions. It is scheduled to begin in November 2023 and end in April 2024.

### Eligibility criteria

**Types of studies.**    All randomised controlled trials of non-pharmacological interventions for postoperative cognitive function will be included. Non-randomised controlled trials, crossover trials, cohort studies, reviews, case reports, expert experience, conference papers, animal studies, and duplicate publications will be excluded.

**Types of participants.**    This study will include patients who underwent surgery under general or epidural anaesthesia, regardless of the surgery type, with no limitations on sex, race, or degree of education. Patients under 18 years of age and those with unstable vital signs will be excluded. To minimise clinical heterogeneity, patients with diagnoses of postoperative delirium, pre-existing cognitive impairment or mental disorders before the surgery, and those with cognitive dysfunction occurring more than 1 year after the surgery will also be excluded [26].

**Types of interventions.**    TEAS, EA, MA, and CT will be included in the intervention group. There will be no restrictions on the duration or timing of these interventions. However, any intervention that involves pharmaceutical treatments aimed at improving cognition will be excluded.

**Types of comparator(s)/control.**    Both the intervention and control groups will undergo anaesthesia and a surgical procedure. The control group will include the no-intervention, routine care, and sham groups. Patients who were undergoing pharmacological interventions associated with improved cognition will be excluded. The sham group will primarily involve sham acupuncture at non-acupoints, sham EA, and sham TEAS. Routine care will include standard postoperative dietary, psychological, and physical care.

**Types of outcomes.**    The primary outcome will be changes in the overall cognitive function from baseline to the end of the intervention; these will be assessed by various neuropsychological scales, including but not limited to the Mini-Mental State Examination (MMSE), Montreal Cognitive Assessment (MoCA), and Alzheimer's Disease Assessment Scale-cognitive subscale (ADAS-cog). The MMSE, a brief pencil-and-paper cognitive test, is the most widely used measure of cognitive function in clinical, research, and community settings [27, 28]. It is well-validated in low-literacy settings [29] and is the primary measure of POCD in existing clinical studies. Compared to the MMSE, the MoCA is a more sensitive test for detecting mild cognitive impairment, especially in the early stages of cognitive decline [30]. The ADAS-cog is a detailed and comprehensive tool that can detect finer changes in cognitive function, which is crucial for monitoring the progression of cognitive decline and the effects of interventions in individuals with Alzheimer's disease or similar conditions [31]. The secondary outcome will be the incidence of POCD, measured based on individual study criteria.

**Table 1. Search strategy for PubMed.**

| No. | Search terms |
|---|---|
| #1 | (postoperat* or *surgery or *surgical or *operative or *operation or anaesthesia). ti, ab |
| #2 | (cognitive or cognition or memory or neurocognit* or intelligenc). ti, ab |
| #3 | #1 AND #2 |
| #4 | postoperative cognitive complications. MeSH |
| #5 | (POCD or postoperative cognitive). ti, ab |
| #6 | #3 OR #4 OR #5 |
| #7 | (cognitive behavioral therapy or acupuncture therapy or electroacupuncture or exercise therapy or music therapy or psychotherapy or neurofeedback). MeSH |
| #8 | (nonpharmacological or nondrug or cognitive behavioral therapy or cognitive therapy or acupuncture or electroacupuncture or acupoint or high-pressure oxygen or biofeedback or rehabilitation or electrical stimulation or training or exercise or music or psychotherapy). ti, ab |
| #9 | #7 OR #8 |
| #10 | (Randomized Controlled Trials as Topic or random allocation or OR clinical study or trial or placebo or random*). ti, ab |
| #11 | (randomized controlled trial or controlled clinical trial or clinical trial) pt |
| #12 | #10 OR #11 |
| #13 | #6 AND #9 AND #12 |

## Search strategy

A systematic search will be conducted using the following electronic databases: PubMed, Embase, Cochrane Library, Web of Science, Cumulative Index to Nursing & Allied Health, Allied and Complementary Medicine Database, and PsycINFO. We will retrieve all English language literature published between January 2000 and November 2023 and screen forward or backward by the references of the included studies. The database search strategy is described in Table 1. The following search terms will be used: (i) disease (POCD), (ii) various non-pharmacological interventions, and (iii) study type (randomized controlled trial). The terms will be used either alone or in combination with "and" and "or."

## Selection of studies

The output of all searches will be imported into Endnote V.X9 software for management. After removing duplicates, two reviewers (Jun Huang and Shanshan Sun) will independently screen titles, keywords, and abstracts for relevance based on the inclusion criteria, excluding ineligible studies. The eligibility of the remaining studies will be determined through full-text review. Any disagreements during the selection process will be resolved through a discussion until a consensus is reached or through consultation with a third author (Kexin Wu). The selection process for this study is illustrated in the flowchart in S1 Fig.

## Data extraction and management

Two investigators (Jun Huang and Shanshan Sun) will independently extract the data using a predesigned extraction form, and any disagreements will be arbitrated by a third senior reviewer (Kexin Wu). The form covers the following items: (1) basic characteristics (first author, year of publication, country), (2) general information (sample size, study design, and distribution ratio), (3) participants (age, sex, type of surgery), (4) intervention characteristics (intervention and control methods, intervention timing, and duration), and (5) results (primary and secondary outcomes). If there are missing data, we will contact the respective corresponding authors.

## Assessment of risk of bias

Two reviewers (Kexin Wu and Jun Huang) will independently assess the methodological quality of selected RCTs using the Risk of Bias Assessment Tool (Rob 2.0) from the Cochrane Collaboration [32]. The tool covers five types of biases: (i) randomization process, (ii) deviation from the intended intervention, (iii) missing outcome data, (iv) result measurement, and (v) selection of reported results. The included trials will be given as having "a low risk of bias," "some concerns," or "a high risk of bias." A third reviewer (Zihan Yin) will be consulted for the final decision-making process.

## Measures of intervention effects

For continuous outcomes, the standard mean difference (SMD) with a 95% confidence interval (CI) will be calculated. For binary outcomes, the collected data will be analysed by calculating the relative risk with 95% CI. Statistical significance of $P < 0.05$ (two-tailed) will be considered statistically significant. If data are presented graphically, the GetData software will be used to obtain them. This software can rapidly extract and transform data from published images. In cases of missing data, the corresponding author will be contacted via email. If data cannot be obtained, only the available data will be analysed.

## Data synthesis

**Pairwise meta-analysis.**   A paired meta-analysis of direct evidence will be conducted using Review Manager V.5.4. We will examine the differences in the participant characteristics, baseline data, intervention measures, and outcomes to assess the presence and degree of clinical heterogeneity across the included studies using $I^2$ statistics and p-values. $I^2$ values exceeding 25%, 50%, and 75% will be considered indicative of low, moderate, and high statistical heterogeneity, respectively. $I^2$ values exceeding 50% will be considered indicative of substantial heterogeneity [33], and a random effects model will be used for the meta-analysis. Otherwise, a fixed effects model will be used to combine the effect sizes.

**Network meta-analysis.**   STATA V.15.1 will be used to generate a network plot showing direct and indirect comparative results, with nodes representing different interventions and lines representing the comparisons made. Larger nodes will correspond to greater sample sizes, while thicker lines will reflect a higher number of RCTS. Each node will represent a different intervention, and the connecting lines between nodes will indicate the effect of comparing the two interventions. Moreover, the Aggregate Data Drug Information System V.1.16.8 will be used to perform a Bayesian NMA based on the Markov chain Monte Carlo algorithm. For all analyses, four chains with 100,000 iterations will be used, discarding an initial burn-in of 20,000 to mitigate the impact of the initial value. A ranking of various non-pharmaceutical interventions will then be generated. The potential scale reduction factor (PSRF) values will be assessed to evaluate the convergence of the results, with a PSRF value close to 1 indicating successful convergence [34].

## Assessment of inconsistency

Consistency is an important principle of NMAs [35] and refers to whether direct and indirect comparisons are consistent. The study will use a node-splitting model to evaluate local inconsistencies at the network level. Consistency assessments will be conducted for each sub-network, followed by a comparison of the outcomes across different sub-networks to ascertain if there are significant differences. Based on the results obtained, a consistent or inconsistent model will be selected. $P < 0.05$ will be considered indicative of a significant difference

between the direct and indirect multiple-treatment comparisons. The results will be interpreted in conjunction with clinical implications and the research background.

## Subgroup analysis and sensitive analysis

Based on the guidelines from Cochrane Handbook 6.1 [36], when $I^2 \geq 50\%$ and $P < 0.05$ (i.e., in case of significant heterogeneity), a subgroup analysis will be performed reasonably according to the intervention timing, type of surgery, and average participant age, among others (if sufficient data are available). Simultaneously, based on the sample size and methodological quality, a sensitivity analysis will be performed by eliminating studies with small sample sizes or a high risk of bias to verify the accuracy and stability of the results.

## Assessment of reporting bias

A comparative adjustment funnel plot will be used to explore the reported bias and small-scale effects [37]. A funnel plot is a simple scatter plot that can be used intuitively to identify publication and other biases. If more than ten studies are included, funnel plots will be used to detect any reported bias. An asymmetrical funnel plot will be considered indicative of publication bias.

## Grading the evidence

We will use the Grading of Recommendations Assessment, Development, and Evaluation (GRADE) [38, 39] system to assess the quality of evidence, which will accordingly be classified as "high," "moderate," "low," or "very low." Two reviewers (Kexin Wu and Jun Huang) will conduct this assessment independently; in case of a disagreement, a third author (Zihan Yin) will be consulted to decide.

## Discussion

Many studies have demonstrated the effectiveness of different non-pharmacological therapies for the prevention and treatment of POCD. However, previous meta-analyses have often been limited to a single non-pharmacological therapy, thereby lacking comprehensive and holistic insights. NMAs emphasize the comparison of multiple interventions under the same conditions, enabling a more thorough and accurate estimation of the treatment effects. Still, there are various non-drug intervention methods, and previous studies have found that TEAS, EA, MA, and CT are used more. Acupuncture has been used to treat a variety of diseases, and EA combines the effects of electrical stimulation with those of conventional MA [40]. TEAS, a combination of transcutaneous electrical nerve stimulation and acupoint therapy, has been found to attenuate brain damage by inhibiting oxidative stress responses [20]. Some studies have found that acupuncture can improve cognitive function by regulating the release of neurotransmitters, reducing the occurrence of neurotoxic effects, and reducing the probability of neuronal degeneration [41]. Cognitive training involves engaging in activities designed to improve cognitive functions, such as memory, attention, and problem-solving. This can be done through computer-based programs, brain games, or other mentally stimulating activities. There remains some confusion amongst clinicians on which interventions are associated with the most positive outcomes.

This study will be the first NMA to compare the effectiveness of four non-pharmacological therapies for POCD. Its results will help generate direct and indirect comparative evidence, ultimately enabling a ranking of these therapies to help clinicians and patients develop intervention plans. To mitigate selection bias, we will adhere to a pre-specified protocol with

explicit inclusion and exclusion criteria and perform a systematic review to ensure transparency and reproducibility. This study will include results from various validated cognitive assessment tools, chosen based on their relevance and sensitivity to detect changes in cognitive function associated with POCD. To minimize bias in the reporting of results, we will attempt to include all pertinent outcomes in the meta-analysis and will contact authors for any missing data as required. Additionally, we will use the GRADE criteria to evaluate the quality of evidence.

This study protocol has some limitations. Firstly, the inclusion of different types of non-pharmacological therapies may introduce a considerable risk of heterogeneity, which may impact our results. Secondly, from the perspective of the study participants, POCD does not have a unified definition. Finally, only English-language trials will be selected, which could result in publication bias.

## Supporting information

**S1 Checklist. PRISMA-P (preferred reporting items for systematic review and meta-analysis protocols) 2015 checklist: Recommended items to address in a systematic review protocol\*.**
(DOC)

**S1 Fig. Preferred reporting items for systematic review and Bayesian network meta-analysis.**
(TIF)

**S1 File. Search strategies of each database.**
(DOCX)

## Acknowledgments

The authors thank all the reviewers for their assistance and support.

## Author Contributions

**Conceptualization:** Qiongnan Bao, Jin Yao, Wanqi Zhong.

**Data curation:** Jun Huang, Shanshan Sun.

**Investigation:** Manze Xia, Zhenghong Chen.

**Methodology:** Kexin Wu, Yaqin Li, Xinyue Zhang.

**Project administration:** Fanrong Liang.

**Software:** Kexin Wu.

**Supervision:** Zihan Yin.

**Writing – original draft:** Kexin Wu, Qiongnan Bao.

**Writing – review & editing:** Zihan Yin, Fanrong Liang.

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
