## [Decision Letter · Decision Letter 0]

13 Feb 2024

PONE-D-23-39876Comparative effectiveness of non-pharmacological therapies for postoperative cognitive dysfunction: Protocol for a systematic review and network meta-analysisPLOS ONE

Dear Dr. Liang,

Thank you for submitting your manuscript to PLOS ONE. After careful consideration, we feel that it has merit but does not fully meet PLOS ONE’s publication criteria as it currently stands. Therefore, we invite you to submit a revised version of the manuscript that addresses the points raised during the review process.

We look forward to receiving your revised manuscript.

Kind regards,

David Chau

Academic Editor

PLOS ONE

Journal Requirements:

"The authors thank the State Administration of Traditional Chinese 

Medicine, the National Natural Science Foundation of China and the 

Science and Technology Department of Sichuan Province for the 

financial support of the project. We also thank editage for editing the 

English text of the draft article."

"This review will be financially supported by the State Administration of Traditional Chinese Medicine, National key research and development program of China (No. 2019YFC1709700), the National Natural Science Foundation of China (Nos. 81973961, 82004486), and the Project of Science and Technology Department of Sichuan Province (No. 2021YFS0087)."

"This review will be financially supported by the State Administration of Traditional Chinese Medicine, National key research and development program of China (No. 2019YFC1709700), the National Natural Science Foundation of China (Nos. 81973961, 82004486), and the Project of Science and Technology Department of Sichuan Province (No. 2021YFS0087)."

6. We notice that your supplementary figures are uploaded with the file type 'Figure'. Please amend the file type to 'Supporting Information'. Please ensure that each Supporting Information file has a legend listed in the manuscript after the references list.

Reviewers' comments:

Reviewer's Responses to Questions

**Comments to the Author**

1. Does the manuscript provide a valid rationale for the proposed study, with clearly identified and justified research questions?

Reviewer #1: No

Reviewer #2: Yes

Reviewer #3: Yes

2. Is the protocol technically sound and planned in a manner that will lead to a meaningful outcome and allow testing the stated hypotheses?

Reviewer #1: No

Reviewer #2: Yes

Reviewer #3: Yes

3. Is the methodology feasible and described in sufficient detail to allow the work to be replicable?

Reviewer #1: No

Reviewer #2: Yes

Reviewer #3: Yes

4. Have the authors described where all data underlying the findings will be made available when the study is complete?

Reviewer #1: Yes

Reviewer #2: Yes

Reviewer #3: Yes

5. Is the manuscript presented in an intelligible fashion and written in standard English?

Reviewer #1: No

Reviewer #2: Yes

Reviewer #3: Yes

6. Review Comments to the Author

You may also provide optional suggestions and comments to authors that they might find helpful in planning their study.

Reviewer #1: 1. You need to define POCD correctly. Do you mean POCD or delirium? Some of the definitions, I think you are talking about delirium and in some of them you are talking about POCD (Delirium is short term (up to 7 days after surgery, POCD occurs later than this). Please decide whether the drive is POCD or delirium and modify / re-write all of your thinking, background and methods accordingly. The mechanistic impact of interventions on these two processes is likely to be different (as are the causes).

2. Why would you include cross-over trials?

3. Types of participants - I would consider stratifying / excluding by surgery severity as there is likely to be no meaningful answer as the impact of different predisposing factors is different. i.e. the things that cause / impact delirium for people having GA hernia repair is different to cardiac surgery etc.... (also what does "undergoing surgery for more than 1 year" mean - I assume this is a typo?)

4. The types of comparator / control need to be more explicit - what is "waiting list" comparator?

5. The primary outcome needs to be POCD o delirium (whatever they pick as the driver for this) and not the MMSE as MMSE may not be recorded by some studies - the phenomena of POCD / delirium will be and therefore they will end up having an incomplete review. (MMSE is also a screening test for cognitive impairment and not delirium).

6. Search strategy should be in appendix - but is very limited and is likely to miss a number of manuscripts.

7. It should like they have already started "The output of all searches was imported into the Endnote V.X9 software for management." and "A comparative adjustment funnel plot was used to explore the

reported bias and small-scale effects" The protocol should be in the future tense.....and ideally they shouldn't have started

8. There is a good deal of the passive tense in the methods - which makes it hard to read.

9. The discussion is extremely short - I know it is a protocol, but I would like some appraisal of the methodology, what definitions they have used, why they chose specific outcomes etc....

Reviewer #2: This study protocol, “Comparative effectiveness of non-pharmacological therapies for postoperative cognitive dysfunction: Protocol for a systematic review and network meta-analysis”, describes meta-analysis methods to evaluate and compare four non-pharmacological interventions to prevent/reduce post-surgical cognitive dysfunction. The value of the study lies in its effort to compare non-pharmacological interventions for a very important and relevant topic. It is a well-written proposal with a robust methodology; however, I have a few comments as mentioned below —

• Pagination and Line Numbers: Include page numbers and line numbers for precise feedback.

• Briefly explain non-pharmacological therapies and how they are similar/different from each other. Additionally, provide insight into their current utilization in the field.

• Demonstrate, how authors plan to control for effects of medications like anticholinergics that have deleterious effects on cognition.

• Additionally, how would you control for the effect of other non-pharmacological treatments like transcranial magnetic brain stimulation that has been shown to improve cognition?

• MMSE is a cognitive screening tool or can be used as a diagnostic adjunct but MMSE is not solely and independently used to make clinical diagnosis of cognitive impairment. Besides, MMSE does not identify early-stage cognitive impairment. Explain how authors plan to address this issue sin their outcomes.

• Briefly explain, especially for new readers, what graphics authors plan to present using the GetData graph digitizer.

• Briefly explain I2 statistics.

• Briefly explain heterogeneity in terms of variability in treatment effects across eligible RCTs.

• Briefly explain the network plot, nodes, lines, and node-splitting model with examples.

• Describe in detail the bias associated with meta-analysis, and how multiple treatment comparison (network) meta-analysis reduces those biases. Explain funnel plot and publication bias.

Reviewer #3: The authors have done an excellent job in crafting a manuscript that is both informative and well-written. The study is described in detail and the language used throughout the article is of the highest quality. The topic being addressed is of great relevance to the scientific community and promises to provide valuable insights upon completion. It's clear that the authors have put a lot of hard work and dedication into this study, and they should be applauded for their efforts. Congratulations on a job well done!

7. PLOS authors have the option to publish the peer review history of their article (what does this mean?). If published, this will include your full peer review and any attached files.

Reviewer #1: **Yes: **Ben Gibbison

Reviewer #2: **Yes: **Sunil Swami

Reviewer #3: **Yes: **Livia Stocco Sanches Valentin

---

## [Author Response · Author response to Decision Letter 0]

29 Mar 2024

Dear Editors and Reviewers:

We sincerely appreciate the editor's and reviewers’ comments concerning our manuscript entitled “Comparative effectiveness of non-pharmacological therapies for postoperative cognitive dysfunction: Protocol for a systematic review and network meta-analysis”. Those comments are all valuable and very helpful for revising and improving our paper, as well as the important guiding significance to our review. We have studied the comments carefully and have made corrections that we hope meet PLOS ONE’s publication criteria. These changes are indicated using text in the red font in the revised manuscript. The main corrections in the paper and the responses to the editor and reviewers’ comments are as follows:

Editor’s Comments: 

David Chau Academic Editor:

Response: Thank you very much for your thoughtful feedback. We have thoroughly reviewed the PLOS ONE style templates you graciously provided and have meticulously revised our manuscript to adhere to the specified requirements. This includes adjustments to file naming, title, authorship details, affiliations, and the main body of the text. We greatly appreciate your guidance in ensuring compliance with the journal's guidelines.

Response: Thank you for your valuable input. We have carefully reviewed and updated the grant information in the 'Funding Information' section as per your suggestion. This study received financial support from the State Administration of Traditional Chinese Medicine, under the National Key Research and Development Program of China (Grant No. 2019YFC1709700), the National Natural Science Foundation of China (Grants Nos. 81973961, 82004486), and the Project of the Science and Technology Department of Sichuan Province (Grant No. 2021YFS0087). Fanrong Liang acknowledges support from these organizations as a grantee.

"The authors thank the State Administration of Traditional Chinese

Medicine, the National Natural Science Foundation of China and the

Science and Technology Department of Sichuan Province for the 

financial support of the project. We also thank editage for editing the 

English text of the draft article."

"This review will be financially supported by the State Administration of Traditional Chinese Medicine, National key research and development program of China (No. 2019YFC1709700), the National Natural Science Foundation of China (Nos. 81973961, 82004486), and the Project of Science and Technology Department of Sichuan Province (No. 2021YFS0087)."

Response: Thanks for your significant comment. We have removed the funding information from the manuscript and added the amended statement in the cover letter. 

"This review will be financially supported by the State Administration of Traditional Chinese Medicine, National key research and development program of China (No. 2019YFC1709700), the National Natural Science Foundation of China (Nos. 81973961, 82004486), and the Project of Science and Technology Department of Sichuan Province (No. 2021YFS0087)."

Response: Thanks for your significant comment. The funders had no role in study design, data collection and analysis, decision to publish, or preparation of the manuscript. We have added the amended Role of Funder statement in the cover letter.

Response: Thanks for your significant recommendation. All authors have decided on a data sharing plan before acceptance.

6. We notice that your supplementary figures are uploaded with the file type 'Figure'. Please amend the file type to 'Supporting Information'. Please ensure that each Supporting Information file has a legend listed in the manuscript after the references list.

Response: Thanks for your suggestion, we have amended the file type to "Supporting Information", and we have listed a legend in the manuscript.

Reviewers' comments:

Reviewer's Responses to Questions

Comments to the Author

1. Does the manuscript provide a valid rationale for the proposed study, with clearly identified and justified research questions?

Reviewer #1: No

Reviewer #2: Yes

Reviewer #3: Yes

2. Is the protocol technically sound and planned in a manner that will lead to a meaningful outcome and allow testing the stated hypotheses?

Reviewer #1: No

Reviewer #2: Yes

Reviewer #3: Yes

3. Is the methodology feasible and described in sufficient detail to allow the work to be replicable?

Reviewer #1: No

Reviewer #2: Yes

Reviewer #3: Yes

4. Have the authors described where all data underlying the findings will be made available when the study is complete?

Reviewer #1: Yes

Reviewer #2: Yes

Reviewer #3: Yes

5. Is the manuscript presented in an intelligible fashion and written in standard English?

Reviewer #1: No

Reviewer #2: Yes

Reviewer #3: Yes

6. Review Comments to the Author

You may also provide optional suggestions and comments to authors that they might find helpful in planning their study.

Reviewer #1:

1.You need to define POCD correctly. Do you mean POCD or delirium? Some of the definitions, I think you are talking about delirium and in some of them you are talking about POCD (Delirium is short term (up to 7 days after surgery, POCD occurs later than this). Please decide whether the drive is POCD or delirium and modify / re-write all of your thinking, background and methods accordingly. The mechanistic impact of interventions on these two processes is likely to be different (as are the causes).

Response: Thank you for your valuable comments. The participants of this study are postoperative cognitive dysfunction (POCD) patients, not postoperative delirium patients. According to the 2018 Naming Consensus on Surgery-Related and Anesthesia-Related Cognitive Injury, ‘postoperative delirium’ is defined as that which occurs in hospital up to 1 week postprocedure or until discharge (whichever occurs first) and meets DSM-5 diagnostic criteria. On the other hand, POCD refers to cognitive impairment present in patients from 30 days to 1 year after surgery.

However, recognizing the disparities between the guidelines' definition and the actual definitions of POCD in RCTs, particularly considering literature predating 2018, which did not explicitly define the timeframe for POCD, we've deliberated on this issue. Consequently, this study will include patients with cognitive decline up to 1 year postoperatively, diagnosed explicitly as POCD in RCTs. Some modifications have been implemented to the introduction and inclusion criteria sections accordingly. Please refer to the manuscript for details. 

Page 5, lines 63-67;

Postoperative cognitive dysfunction (POCD) is a common postoperative complication of the central nervous system and is defined as cognitive decline occurring within 1 year after a surgery (1, 2). This decline is characterized by a decrease in memory, abstraction, and attention (3). 

Page 9, lines 135-143;

This study will include patients who underwent surgery under general or epidural anaesthesia, regardless of the surgery type, with no limitations on the participant's sex, race, or degree of education. We will exclude patients aged < 18 years and those with unstable vital signs. To reduce clinical heterogeneity, we will also exclude patients with diagnoses of postoperative delirium, pre-existing cognitive impairment or mental disorders before the surgery, and cognitive dysfunction occurring more than 1 year after the surgery (27). 

2. Why would you include cross-over trials?

Response: Thank you for your feedback. Initially, our plan included crossover trials with Phase I outcomes available, as the initial part of the study is influenced by a single intervention/control method, and results after crossover intervention would not be included. However, upon further consideration and taking into account your suggestion, and also considering that no existing crossover trials on POCD were found in our actual search process, so we have excluded crossover trials from our study. Please refer to the manuscript for details. 

Page 8, lines 129-134;

All randomised controlled trials of non-pharmacological interventions for postoperative cognitive function will be included. Non-randomised controlled trials, crossover trials, cohort studies, reviews, case reports, expert experience, conference papers, animal studies, and duplicate publications will be excluded.

3. Types of participants - I would consider stratifying/excluding by surgery severity as there is likely to be no meaningful answer as the impact of different predisposing factors is different. i.e. the things that cause / impact delirium for people having GA hernia repair is different to cardiac surgery etc.... (also what does "undergoing surgery for more than 1 year" mean - I assume this is a typo?)

Response: Thank you for your suggestion. As you pointed out, there may be no meaningful answer for different surgical severity because of the different effects of different preconditions. Therefore, in this study, we will conduct a reasonable subgroup analysis according to the intervention timing, type of surgery, average age of the participant, and so on. 

We have decided not to include a category for "more than 1 year after surgery" as POCD is defined as cognitive impairment present during the 1-year post-surgery period. Cognitive impairment occurring more than 1 year post-surgery may be considered mild cognitive impairment rather than POCD. Please refer to the manuscript for details. 

Page 17, lines 253-261;

Based on the guidelines from Cochrane Handbook 6.1 (35), when I2 ≥50% and P <0.05 (i.e., in case of significant heterogeneity), a subgroup analysis will be performed reasonably according to the intervention timing, type of surgery, and average participant age, among others (if sufficient data are available). Simultaneously, based on the sample size and methodological quality, a sensitivity analysis will be performed by eliminating studies with small sample sizes or a high risk of bias to verify the accuracy and stability of the results.

Page 9, lines 135-143;

This study will include patients who underwent surgery under general or epidural anaesthesia, regardless of the surgery type, with no limitations on the participant's sex, race, or degree of education. We will exclude patients aged < 18 years and those with unstable vital signs. To reduce clinical heterogeneity, we will also exclude patients with diagnoses of postoperative delirium, pre-existing cognitive impairment or mental disorders before the surgery, and cognitive dysfunction occurring more than 1 year after the surgery (27). 

4. The types of comparator / control need to be more explicit - what is "waiting list" comparator?

Response: Thank you for your suggestion. Considering the current RCT studies, due to the nature of postoperative conditions, all groups receive standard care and treatment, with no specific distinction made between standard care and the non-intervention group. After discussion, we have decided to define the comparison groups/control groups as follows: a. Non-intervention group; b. Routine care group; c. Sham group. We have made some modifications to the inclusion criteria for the control group. Please refer to the manuscript for details.

Pages 9-10

---

## [Decision Letter · Decision Letter 1]

26 Apr 2024

PONE-D-23-39876R1Comparative effectiveness of non-pharmacological therapies for postoperative cognitive dysfunction: Protocol for a systematic review and network meta-analysisPLOS ONE

Dear Dr. Liang,

Thank you for submitting your manuscript to PLOS ONE. After careful consideration, we feel that it has merit but does not fully meet PLOS ONE’s publication criteria as it currently stands. Therefore, we invite you to submit a revised version of the manuscript that addresses the points raised during the review process. specifically, reviewer-1 considers a start of search and review of your protocol that goes against the prerequests of protocol submission:https://journals.plos.org/plosone/s/reviewer-guidelines#loc-study-protocols

We look forward to receiving your revised manuscript.

Kind regards,

David Chau

Academic Editor

PLOS ONE

Journal Requirements:

Reviewers' comments:

Reviewer's Responses to Questions

**Comments to the Author**

1. Does the manuscript provide a valid rationale for the proposed study, with clearly identified and justified research questions?

Reviewer #1: No

Reviewer #2: Yes

Reviewer #3: Yes

2. Is the protocol technically sound and planned in a manner that will lead to a meaningful outcome and allow testing the stated hypotheses?

Reviewer #1: No

Reviewer #2: Yes

Reviewer #3: Yes

3. Is the methodology feasible and described in sufficient detail to allow the work to be replicable?

Reviewer #1: No

Reviewer #2: Yes

Reviewer #3: Yes

4. Have the authors described where all data underlying the findings will be made available when the study is complete?

Reviewer #1: Yes

Reviewer #2: Yes

Reviewer #3: Yes

5. Is the manuscript presented in an intelligible fashion and written in standard English?

Reviewer #1: No

Reviewer #2: Yes

Reviewer #3: Yes

6. Review Comments to the Author

You may also provide optional suggestions and comments to authors that they might find helpful in planning their study.

Reviewer #1: The authors have already started the search and review process - therefore there is little point in reviewing a protocol to which the authors have already started (see point 7 in 1st review) and their response to point 2 in this review in which they say they have actually started searching and so they have excluded cross - over studies.

You can't exclude patients with a diagnosis of delirium in your exclusion criteria as delirium is closely associated with cognitive dysfunction. You have to choose your outcome measure that means you are measuring POCD and not delirium

You cannot review a protocol where the work has already started as there is little point in providing any feedback.

Reviewer #2: Overall, this manuscript is well-written and includes a good foundation of information and methodology. To improve the paper further, I recommend adding more details, particularly in the methodology and discussion sections. For example, acupuncture, cognitive training, and transcutaneous electrical acupoint stimulation are mentioned in the intervention types but there are no additional details about them even in the discussion section. Furthermore, the manuscript talks about routine care and sham groups but provides no additional details. It would be helpful to discuss the different types of biases associated with meta-analysis and provide reasoning and remedies. Additionally, something like this "The study will use a node-splitting model to evaluate local inconsistencies at the network level." will not resonate with a lot of readers; a little more explanation will help them conceptualize the method and better understand the analysis. Expanding on areas like these will enhance the clarity and depth of your work, making it more accessible to a wider audience and offering greater insight into the findings.

Reviewer #3: I have observed significant improvements in the manuscript's structure and the conceptualization of terms related to cognitive dysfunction. I am confident that this paper merits consideration for publication in one of the editions of the esteemed PlosOne journal.

7. PLOS authors have the option to publish the peer review history of their article (what does this mean?). If published, this will include your full peer review and any attached files.

Reviewer #1: **Yes: **Ben Gibbison

Reviewer #2: **Yes: **SUNIL SWAMI

Reviewer #3: **Yes: **Livia Stocco Sanches Valentin

---

## [Author Response · Author response to Decision Letter 1]

17 Jun 2024

Editor’s Comments: 

David Chau Academic Editor:

1. Therefore, we invite you to submit a revised version of the manuscript that addresses the points raised during the review process.

specifically, reviewer-1 considers a start of search and review of your protocol that goes against the prerequests of protocol submission:

https://journals.plos.org/plosone/s/reviewer-guidelines#loc-study-protocols

Response: Thank you very much for your thoughtful feedback. We have meticulously revised our manuscript to adhere to the specified requirements. n response to reviewer 1's suggestion, we need to make the necessary explanations： The study has not officially begun, but a pre-search of the topics has been conducted to develop a research protocol. We sincerely apologize for any misunderstandings arising from the misuse of tenses in the initial draft. We are committed to embracing the reviewer's recommendations with an open mind and we greatly appreciate your guidance in ensuring compliance with the journal's guidelines.

Reviewers' comments:

Reviewer's Responses to Questions

Comments to the Author

1. Does the manuscript provide a valid rationale for the proposed study, with clearly identified and justified research questions?

Reviewer #1: No

Reviewer #2: Yes

Reviewer #3: Yes

2. Is the protocol technically sound and planned in a manner that will lead to a meaningful outcome and allow testing the stated hypotheses?

Reviewer #1: No

Reviewer #2: Yes

Reviewer #3: Yes

3. Is the methodology feasible and described in sufficient detail to allow the work to be replicable?

Reviewer #1: No

Reviewer #2: Yes

Reviewer #3: Yes

4. Have the authors described where all data underlying the findings will be made available when the study is complete?

Reviewer #1: Yes

Reviewer #2: Yes

Reviewer #3: Yes

5. Is the manuscript presented in an intelligible fashion and written in standard English?

Reviewer #1: No

Reviewer #2: Yes

Reviewer #3: Yes

6. Review Comments to the Author Please use the space provided to explain your answers to the questions above and, if applicable, provide comments about issues authors must address before this protocol can be accepted for publication. You may also include additional comments for the author, including concerns about research or publication ethics.

You may also provide optional suggestions and comments to authors that they might find helpful in planning their study.

Reviewer #1: The authors have already started the search and review process - therefore there is little point in reviewing a protocol to which the authors have already started (see point 7 in 1st review) and their response to point 2 in this review in which they say they have actually started searching and so they have excluded cross - over studies.

You can't exclude patients with a diagnosis of delirium in your exclusion criteria as delirium is closely associated with cognitive dysfunction. You have to choose your outcome measure that means you are measuring POCD and not delirium

You cannot review a protocol where the work has already started as there is little point in providing any feedback.

Response: Thank you for your feedback. This study has conducted a preliminary search to determine the retrieval formula. The research has not yet started, so we will also sincerely learn and accept your opinions. As you said, postoperative delirium (POD) is closely related to postoperative cognitive impairment. POD is defined as the clinical syndrome of transient fluctuating disturbance in attention, mental status, and consciousness, occurring immediately postoperatively after anesthesia and surgery up to during hospitalization(1, 2). Unlike POD being diagnosed with clinical symptoms, POCD is defined by neuropsychological tests including different domains of impaired cognition, such as verbal memory, visual memory, language comprehension, visuospatial abstraction, attention, and concentration(3). Before 2018, the standard for POCD was limited to research. There was no standardized definition.(4). In this study, acknowledging the availability of well-defined neuropsychological scales for the delineation and diagnosis of POCD, as well as the established diagnostic criteria for POCD within the framework of guideline definitions and RCTs, the Mini-Mental State Examination (MMSE) was selected as the primary outcome measure. Concurrently, the incidence rate of POCD was designated as the secondary outcome measure.

1. Newman MF, Kirchner JL, Phillips-Bute B, Gaver V, Grocott H, Jones RH, et al. Longitudinal assessment of neurocognitive function after coronary-artery bypass surgery. The New England Journal of Medicine. 2001;344(6):395-402.

2. Robinson TN, Raeburn CD, Tran ZV, Angles EM, Brenner LA, Moss M. Postoperative delirium in the elderly: risk factors and outcomes. Ann Surg. 2009;249(1):173-8. http://doi.org/10.1097/SLA.0b013e31818e4776

3. Moller JT, Cluitmans P, Rasmussen LS, Houx P, Rasmussen H, Canet J, et al. Long-term postoperative cognitive dysfunction in the elderly ISPOCD1 study. ISPOCD investigators. International Study of Post-Operative Cognitive Dysfunction. Lancet (London, England). 1998;351(9106):857-61.

4. Evered L, Silbert B, Knopman DS, Scott DA, DeKosky ST, Rasmussen LS, et al. Recommendations for the Nomenclature of Cognitive Change Associated with Anaesthesia and Surgery-2018. Anesthesiology. 2018;129(5):872-9. http://doi.org/10.1097/ALN.0000000000002334

Reviewer #2: Overall, this manuscript is well-written and includes a good foundation of information and methodology. To improve the paper further, I recommend adding more details, particularly in the methodology and discussion sections. For example, acupuncture, cognitive training, and transcutaneous electrical acupoint stimulation are mentioned in the intervention types but there are no additional details about them even in the discussion section. Furthermore, the manuscript talks about routine care and sham groups but provides no additional details. It would be helpful to discuss the different types of biases associated with meta-analysis and provide reasoning and remedies. Additionally, something like this "The study will use a node-splitting model to evaluate local inconsistencies at the network level." will not resonate with a lot of readers; a little more explanation will help them conceptualize the method and better understand the analysis. Expanding on areas like these will enhance the clarity and depth of your work, making it more accessible to a wider audience and offering greater insight into the findings.

Response: We extend our heartfelt gratitude for the affirmation and constructive feedback provided. In deference to your suggestions, we have implemented the subsequent revisions to the manuscript.

(1) We further elaborate on intervention methods (MA, EA, TEAS, and CT) and control methods (usual care and sham) in the Methods and Discussion section. Page 9, lines 150-157; Page 19, lines 291-306；

(2) The protocol will explore the sources of bias risk through subgroup analysis, sensitivity analysis and other methods. Page 20, lines 311-316；

(3) The protocol has been elaborated on the methodological details of "Assessment of Inconsistency," which also provides more specific guidance for our subsequent work. Page 17, lines 253-259.

Reviewer #3: I have observed significant improvements in the manuscript's structure and the conceptualization of terms related to cognitive dysfunction. I am confident that this paper merits consideration for publication in one of the editions of the esteemed PlosOne journal.

Response: Thank you very much for your affirmation and feedback.

---

## [Decision Letter · Decision Letter 2]

3 Jul 2024

PONE-D-23-39876R2Comparative effectiveness of non-pharmacological therapies for postoperative cognitive dysfunction: Protocol for a systematic review and network meta-analysisPLOS ONE

Dear Dr. Liang,

Thank you for submitting your manuscript to PLOS ONE. After careful consideration, we feel that it has merit but does not fully meet PLOS ONE’s publication criteria as it currently stands. Therefore, we invite you to submit a revised version of the manuscript that addresses the points raised during the review process.

We look forward to receiving your revised manuscript.

Kind regards,

David Chau

Academic Editor

PLOS ONE

Journal Requirements:

Reviewers' comments:

Reviewer's Responses to Questions

**Comments to the Author**

1. Does the manuscript provide a valid rationale for the proposed study, with clearly identified and justified research questions?

Reviewer #1: Yes

Reviewer #2: Yes

2. Is the protocol technically sound and planned in a manner that will lead to a meaningful outcome and allow testing the stated hypotheses?

Reviewer #1: No

Reviewer #2: Yes

3. Is the methodology feasible and described in sufficient detail to allow the work to be replicable?

Reviewer #1: Yes

Reviewer #2: Yes

4. Have the authors described where all data underlying the findings will be made available when the study is complete?

Reviewer #1: Yes

Reviewer #2: Yes

5. Is the manuscript presented in an intelligible fashion and written in standard English?

Reviewer #1: No

Reviewer #2: Yes

6. Review Comments to the Author

You may also provide optional suggestions and comments to authors that they might find helpful in planning their study.

Reviewer #1: I think the manuscript needs some language editing as part of this is in the wrong tense and it makes me unsure whether the work has started, some of it reads like a trial.... so the authors need to be very specific about what they mean. I do not want to punish people for writing in their second language - but tenses change the meaning of words in this protocol and it makes me uncertain as to the quality of the end product.

I also think that your primary outcomes need to not be about MMSE - but more about change from the mean (i.e. mean difference) in cognitive scores. This would allow you to use outcomes where they have not used MMSE, but another (and arguably better cognitive score - e.g. MOCA). Primary outcome should be generic - i.e. change in cognitive score etc.

Reviewer #2: Dear Authors, I enjoyed reading this paper. The proposed literature research and the analysis will provide valuable insights into the topic and will contributed to the field. Thank you for your good work!

7. PLOS authors have the option to publish the peer review history of their article (what does this mean?). If published, this will include your full peer review and any attached files.

Reviewer #1: **Yes: **Ben Gibbison

Reviewer #2: No

---

## [Author Response · Author response to Decision Letter 2]

15 Jul 2024

Dear Editors and Reviewers:

We sincerely appreciate the editor's and reviewers’ comments concerning our manuscript entitled “Comparative effectiveness of non-pharmacological therapies for postoperative cognitive dysfunction: Protocol for a systematic review and network meta-analysis”. Those comments are all valuable and very helpful for revising and improving our paper, as well as the important guiding significance to our review. We have studied the comments carefully and have made corrections that we hope meet PLOS ONE’s publication criteria. These changes are indicated using text in the red font in the revised manuscript. The main corrections in the paper and the responses to the editor and reviewers’ comments are as follows:

Editor’s Comments: 

David Chau Academic Editor:

1. Thank you for submitting your manuscript to PLOS ONE. After careful consideration, we feel that it has merit but does not fully meet PLOS ONE’s publication criteria as it currently stands. Therefore, we invite you to submit a revised version of the manuscript that addresses the points raised during the review process.

Response: We are deeply grateful for your thoughtful feedback. We have meticulously revised our manuscript once again to meet the stipulated requirements. We are always committed to embracing your and the reviewers' suggestions with an open mind, and we greatly appreciate your guidance in ensuring compliance with the journal's guidelines. For the reference list of this study, we have once again ensured that it is complete and correct.

Reviewers' comments:

Reviewer's Responses to Questions

Comments to the Author

1. Does the manuscript provide a valid rationale for the proposed study, with clearly identified and justified research questions?

Reviewer #1: Yes

Reviewer #2: Yes

2. Is the protocol technically sound and planned in a manner that will lead to a meaningful outcome and allow testing the stated hypotheses?

Reviewer #1: No

Reviewer #2: Yes

3. Is the methodology feasible and described in sufficient detail to allow the work to be replicable?

Reviewer #1: Yes

Reviewer #2: Yes

4. Have the authors described where all data underlying the findings will be made available when the study is complete?

Reviewer #1: Yes

Reviewer #2: Yes

5. Is the manuscript presented in an intelligible fashion and written in standard English?

Reviewer #1: No

Reviewer #2: Yes

6. Review Comments to the Author

You may also provide optional suggestions and comments to authors that they might find helpful in planning their study.

Reviewer #1: I think the manuscript needs some language editing as part of this is in the wrong tense and it makes me unsure whether the work has started, some of it reads like a trial.... so the authors need to be very specific about what they mean. I do not want to punish people for writing in their second language - but tenses change the meaning of words in this protocol and it makes me uncertain as to the quality of the end product.

I also think that your primary outcomes need to not be about MMSE - but more about change from the mean (i.e. mean difference) in cognitive scores. This would allow you to use outcomes where they have not used MMSE, but another (and arguably better cognitive score - e.g. MOCA). Primary outcome should be generic - i.e. change in cognitive score etc.

Response: Thank you for your insightful comments and for bringing the tense issues to our attention. We have taken your feedback very seriously and have made the necessary revisions to the manuscript. The tense has been corrected throughout the document to ensure clarity and consistency. We are grateful for the opportunity to refine our work and hope that it now meets the journal's standards.

Additionally, your suggestion to focus the primary outcomes of our study on change from the mean (i.e. mean difference) in cognitive scores, rather than on the MMSE, has prompted us to consider more deeply the connection between outcomes and cognitive function. However, we would like to clarify the reasons for using the MMSE in this study: (1) The MMSE is a widely recognised and standardised tool in the field of neuropsychological assessment, facilitating easy comparison with a vast body of existing literature. The simplicity of its administration and scoring makes it accessible to a broad range of healthcare providers and settings, which is crucial for our study. (2) Based on our preliminary search for RCTs related to non-pharmacological therapies for POCD, the MMSE is the most frequently used tool compared to other neurocognitive tests, such as the MoCA. However, as the limited number of studies using MoCA may not be sufficient for a network meta-analysis, the extensive use of MMSE ensures that our results are relevant and applicable to a wider reader. (3) We aim to focus on changes in MMSE scores rather than changes in the mean (i.e. mean difference) of cognitive scores because MoCA and the MMSE are inherently different metrics; unifying the two into a single measure is controversial. By focusing on the MMSE, we aim to maintain consistency in data collection which is vital for the reliability and validity of our research findings. 

However, we will consider your suggestion and, where data are available, utilize the MoCA as a secondary outcome measure to perform additional analyses on cognitive changes, thereby providing a broader interpretation of our results. We are open to further discussion and are willing to make revisions to enhance the contribution of our study to the field.

Reviewer #2: Dear Authors, I enjoyed reading this paper. The proposed literature research and the analysis will provide valuable insights into the topic and will contributed to the field. Thank you for your good work!

Response: Thank you for your positive and encouraging feedback on our manuscript. We are pleased to hear that our research is considered valuable to the field. We appreciate your time and the interest you have shown in our work.

---

## [Decision Letter · Decision Letter 3]

2 Aug 2024

PONE-D-23-39876R3Comparative effectiveness of non-pharmacological therapies for postoperative cognitive dysfunction: Protocol for a systematic review and network meta-analysisPLOS ONE

Dear Dr. Liang,

Thank you for submitting your manuscript to PLOS ONE. After careful consideration, we feel that it has merit but does not fully meet PLOS ONE’s publication criteria as it currently stands. Therefore, we invite you to submit a revised version of the manuscript that addresses the points raised during the review process.

We look forward to receiving your revised manuscript.

Kind regards,

David Chau

Academic Editor

PLOS ONE

Journal Requirements:

Reviewers' comments:

Reviewer's Responses to Questions

**Comments to the Author**

1. Does the manuscript provide a valid rationale for the proposed study, with clearly identified and justified research questions?

Reviewer #1: Yes

2. Is the protocol technically sound and planned in a manner that will lead to a meaningful outcome and allow testing the stated hypotheses?

Reviewer #1: No

3. Is the methodology feasible and described in sufficient detail to allow the work to be replicable?

Reviewer #1: Yes

4. Have the authors described where all data underlying the findings will be made available when the study is complete?

Reviewer #1: Yes

5. Is the manuscript presented in an intelligible fashion and written in standard English?

Reviewer #1: Yes

6. Review Comments to the Author

You may also provide optional suggestions and comments to authors that they might find helpful in planning their study.

Reviewer #1: I still don't think you should use a single tool as your outcome as it will exclude too many studies. It will effectively create selection bias of the manuscripts you include.

You should include outcomes from all tools / instruments and present the outcome as change from the central tendency (median / mean).

7. PLOS authors have the option to publish the peer review history of their article (what does this mean?). If published, this will include your full peer review and any attached files.

Reviewer #1: No

---

## [Author Response · Author response to Decision Letter 3]

11 Aug 2024

Dear Editors and Reviewers:

We sincerely appreciate the editor's and reviewers’ comments concerning our manuscript entitled “Comparative effectiveness of non-pharmacological therapies for postoperative cognitive dysfunction: Protocol for a systematic review and network meta-analysis”. Those comments are all valuable and very helpful for revising and improving our paper, as well as the important guiding significance to our review. We have studied the comments carefully and have made corrections that we hope meet PLOS ONE’s publication criteria. These changes are indicated using text in the red font in the revised manuscript. The main corrections in the paper and the responses to the editor and reviewers’ comments are as follows:

Editor’s Comments: 

David Chau Academic Editor:

1. Thank you for submitting your manuscript to PLOS ONE. After careful consideration, we feel that it has merit but does not fully meet PLOS ONE’s publication criteria as it currently stands. Therefore, we invite you to submit a revised version of the manuscript that addresses the points raised during the review process.

Response: We are deeply grateful for your thoughtful feedback. We have meticulously revised our manuscript once again to meet the stipulated requirements. We are always committed to embracing your and the reviewers' suggestions with an open mind, and we greatly appreciate your guidance in ensuring compliance with the journal's guidelines. For the reference list of this study, we have once again ensured that it is complete and correct.

Reviewers' comments:

Reviewer's Responses to Questions

Comments to the Author

1. Does the manuscript provide a valid rationale for the proposed study, with clearly identified and justified research questions?

Reviewer #1: Yes

2. Is the protocol technically sound and planned in a manner that will lead to a meaningful outcome and allow testing the stated hypotheses?

Reviewer #1: No

3. Is the methodology feasible and described in sufficient detail to allow the work to be replicable?

Reviewer #1: Yes

4. Have the authors described where all data underlying the findings will be made available when the study is complete?

Reviewer #1: Yes

5. Is the manuscript presented in an intelligible fashion and written in standard English?

Reviewer #1: Yes

6. Review Comments to the Author

You may also provide optional suggestions and comments to authors that they might find helpful in planning their study.

Reviewer #1: I still don't think you should use a single tool as your outcome as it will exclude too many studies. It will effectively create selection bias of the manuscripts you include.

You should include outcomes from all tools/instruments and present the outcome as change from the central tendency (median/mean).

Response: We extend our sincerest gratitude for your insightful comments and suggestions on our manuscript. Upon careful consideration, we recognize that limiting the main outcome indicator to a single cognitive assessment tool may limit the breadth and depth of our research. Accordingly, we have adjusted our primary outcome to encompass the change in cognitive scores from a variety of cognitive assessment tools, including not only the MMSE but also the MoCA, ADAS-Cog, etc. This amendment ensures that our study is not constrained by the use of a single assessment instrument, thereby mitigating the risk of selection bias and allowing for a more comprehensive analysis of cognitive changes across different studies.

In our statistical analysis, we will employ the standardized mean difference (SMD) to quantify the changes in continuous variables, which will offer a standardized metric for comparing the magnitude of changes across different cognitive assessment scales.

We trust that these revisions address your concerns and enhance the quality and relevance of our research. We look forward to your continued guidance and feedback.

---

## [Decision Letter · Decision Letter 4]

15 Aug 2024

Comparative effectiveness of non-pharmacological therapies for postoperative cognitive dysfunction: Protocol for a systematic review and network meta-analysis

PONE-D-23-39876R4

Dear Dr. Liang,

We’re pleased to inform you that your manuscript has been judged scientifically suitable for publication and will be formally accepted for publication once it meets all outstanding technical requirements.

Kind regards,

David Chau

Academic Editor

PLOS ONE

Additional Editor Comments (optional):

Reviewers' comments:

Reviewer's Responses to Questions

**Comments to the Author**

1. Does the manuscript provide a valid rationale for the proposed study, with clearly identified and justified research questions?

Reviewer #1: Yes

2. Is the protocol technically sound and planned in a manner that will lead to a meaningful outcome and allow testing the stated hypotheses?

Reviewer #1: Partly

3. Is the methodology feasible and described in sufficient detail to allow the work to be replicable?

Reviewer #1: Yes

4. Have the authors described where all data underlying the findings will be made available when the study is complete?

Reviewer #1: Yes

5. Is the manuscript presented in an intelligible fashion and written in standard English?

Reviewer #1: Yes

6. Review Comments to the Author

You may also provide optional suggestions and comments to authors that they might find helpful in planning their study.

Reviewer #1: The authors have answered this set of comments. I am still not certain that they have not started the review…..

7. PLOS authors have the option to publish the peer review history of their article (what does this mean?). If published, this will include your full peer review and any attached files.

Reviewer #1: **Yes: **Ben Gibbison

---

## [Editor Report · Acceptance letter]

28 Aug 2024

PONE-D-23-39876R4 

PLOS ONE

Dear Dr. Liang, 

I'm pleased to inform you that your manuscript has been deemed suitable for publication in PLOS ONE. Congratulations! Your manuscript is now being handed over to our production team.

Kind regards, 

on behalf of

Dr. David Chau 

Academic Editor

PLOS ONE